# Assessment of Changes in Physiological Markers in Different Body Fluids at Rest and after Exercise

**DOI:** 10.3390/nu14214685

**Published:** 2022-11-05

**Authors:** Amalini Jesuthasan, Ajmol Ali, Jason Kai Wei Lee, Kay Rutherfurd-Markwick

**Affiliations:** 1School of Health Sciences, Massey University, Auckland 0745, New Zealand; 2School of Sport, Exercise and Nutrition, Massey University, Auckland 0745, New Zealand; 3Centre for Metabolic Health Research, Massey University, Auckland 0745, New Zealand; 4Heat Resilience and Performance Centre, Yong Loo Lin School of Medicine, National University of Singapore, Singapore 119228, Singapore; 5Human Potential Translational Research Programme, Yong Loo Lin School of Medicine, National University of Singapore, Singapore 119228, Singapore; 6Department of Physiology, Yong Loo Lin School of Medicine, National University of Singapore, Singapore 117593, Singapore; 7Campus for Research Excellence and Technological Enterprise (CREATE), 1 CREATE Way, Singapore 138602, Singapore

**Keywords:** blood, urine, saliva, sweat, sports, performance, nutrition, health

## Abstract

Physiological and biological markers in different body fluids are used to measure the body’s physiological or pathological status. In the field of sports and exercise medicine, the use of these markers has recently become more popular for monitoring an athlete’s training response and assessing the immediate or long-term effects of exercise. Although the effect of exercise on different physiological markers using various body fluids is well substantiated, no article has undertaken a review across multiple body fluids such as blood, saliva, urine and sweat. This narrative review aims to assess various physiological markers in blood, urine and saliva, at rest and after exercise and examines physiological marker levels obtained across similar studies, with a focus on the population and study methodology used. Literature searches were conducted using PRISMA guidelines for keywords such as exercise, physical activity, serum, sweat, urine, and biomarkers, resulting in an analysis of 15 studies for this review paper. When comparing the effects of exercise on physiological markers across different body fluids (blood, urine, and saliva), the changes detected were generally in the same direction. However, the extent of the change varied, potentially as a result of the type and duration of exercise, the sample population and subject numbers, fitness levels, and/or dietary intake. In addition, none of the studies used solely female participants; instead, including males only or both male and female subjects together. The results of some physiological markers are sex-dependent. Therefore, to better understand how the levels of these biomarkers change in relation to exercise and performance, the sex of the participants should also be taken into consideration.

## 1. Introduction

The adult human body is made up of 60% water by mass, and body fluids play an important regulatory role in homeostasis [1]. Total body fluids can be categorised into extracellular fluid (ECF) and intracellular fluid (ICF), which includes all the fluid enclosed in cells by their plasma membrane. Most chemical reactions occur in the ICF, and this accounts for 40% of total body weight. Of the remaining total body weight, 20% is contained within ECF, and this can be further divided into plasma and interstitial fluid (IF). Body fluids vary in composition between body compartments, depending upon exchanges between the cells in the biological tissues and blood. For example, sodium and chloride are present in high amounts in the ECF (plasma), whereas potassium and phosphate are higher in the ICF [2]. Changes in the concentration or composition of biochemical constituents in body fluids are therefore used as indicators of physiological or pathological conditions [3]. However, the composition of some body fluids is useful only at certain time points, which limits their usefulness; for example, measures of urine specific gravity are a useful measure of hydration status prior to exercise but not post-exercise [4]. Saliva shows large variability in flow rate during low levels of hydration [5], as well as variation in results due to interference with food and beverage consumption, oral hygiene routines, dental erosions and circadian rhythm [6].

A physiological or biological marker is a measurable indicator of the body’s physiological or pathological status. Plasma, serum, urine, saliva, sweat, stool, cerebrospinal liquid, tears, and amniotic fluid are used for diagnostic purposes as well as in scientific and health research; of these, serum, plasma, urine, and saliva are the most popular due to the ease of collection, lower ethical cost and the wide range of tests that can be performed using these fluids [7]. In the field of sports and exercise medicine, the use of physiological markers has become a popular method for monitoring an athlete’s training response, as the body’s physiology is significantly different at rest and after exercise [8,9,10,11,12]. These markers may be used to assess the immediate or long-term effects of exercise. In sports and exercise, body fluids such as urine and saliva are preferred, as the collection is simple, non-invasive, cheap and does not require a specialised technician [13,14]. A physiological marker’s concentration during exercise depends on several variables, including, but not limited to, the length and intensity of exercise, degree of exhaustion and hydration status of the athlete, as well as age, sex and fitness level [15].

Compared to the resting state, exercise poses a substantial increase in demand on the body and disturbs the body’s homeostasis, causing a shift in hemodynamic and metabolic processes, which leads to both fluid and electrolyte changes in the body [16]. Although the effect of exercise on various body fluids is well substantiated, to our knowledge, no article has undertaken a review across multiple body fluids. Therefore, the purpose of this article is to assess various physiological markers in different body fluids, namely blood, urine and saliva, at rest and after exercise. This review also aims to compare the results obtained across studies measuring physiological markers pre and post-exercise and suggest possible reasons for the variations in results obtained, with a focus on the population and sample collection or methodology used. In addition, where possible, the change in (physiological marker) level due to the same or similar exercise protocol between different body fluids will be compared.

## 2. Materials and Methods

Comprehensive narrative literature searches were conducted between August 2020 to June 2021 using Google Scholar, Scopus and PUBMED. Keywords such as exercise, physical activity, serum, sweat, urine, and biomarkers, along with Boolean operators such as ‘and’ or ‘or’, were used. Results were screened for relevance for this publication and excluded if the studies included any supplementation intervention. Studies that reported the actual value of physiological markers before and after exercise, rather than a trend or change, were used to compile the Tables. Figure 1 shows the PRISMA flow diagram of the search procedure used for this narrative review.

## 3. Results and Discussion

### 3.1. Blood (Serum and Plasma)

Blood is a vital fluid required for the transport of nutrients, gases and waste; it also serves to protect against infection by transporting white blood cells to target pathogens, control pH and temperature and aid with clotting at a site of injury [17]. Blood is a complex mixture of plasma, white blood cells, red blood cells and platelets. Plasma contains about 92% water, with the remainder comprising dissolved gases, proteins such as albumin and globulin, immunoglobulins and electrolytes such as sodium, potassium, bicarbonate, chloride and calcium [18]. Both blood plasma and serum are widely used in research, and many investigators consider them to be similar; however, they are not, and the inappropriate use of plasma or serum during diagnostic testing can lead to an incorrect diagnosis [19]. The serum is obtained from blood that has coagulated and, as a result, the fibrin clot, blood cells and associated coagulation factors produced during coagulation are separated from the serum following centrifugation [20]. Plasma samples are obtained when samples are collected in the presence of an anticoagulant (such as EDTA or heparin) and centrifuged to extract cellular components [21]. Plasma has a slightly higher protein concentration than serum, with average values of 74.5 g/L–75.8 g/L for heparinised plasma [22] versus 72.1 g/L–72.9 g/L for serum [23]. This difference has been attributed to the conversion of fibrinogen to fibrin during in vitro clot formation in serum [22].

During exercise, blood ensures the transport of oxygen from the lungs to tissues, and the delivery of metabolically produced carbon dioxide to the lungs for expiration [24]. While exercising, sweat-induced hypohydration reduces plasma volume and increases plasma osmotic pressure in proportion to the amount of fluid lost [25]. Plasma volume reduces because it provides the precursor fluid for sweat, and osmolality increases because sweat is ordinarily hypotonic relative to plasma. Sodium and chloride are primarily responsible for the increased plasma osmolality. Plasma volume expansion normally occurs both with acute endurance exercise and training, with an 11.6% increase in plasma shown to be induced after three days of high-intensity exercise [26]. Plasma-albumin labelling methods and Evan’s blue-dye analysis are the most accurate ways to measure plasma volumes in steady-state [27]. These protein-based techniques, however, assume that protein content will remain constant throughout the measurement time, yet, proteins frequently change while plasma volume is rapidly changing, so these approaches cannot be used when plasma volume is changing. An alternative way to estimate changes in plasma volume is to measure hematocrit or hemoglobin changes [28]. Hemodilution caused by plasma volume expansion leads to a decrease in the hemoglobin concentration and hematocrit and is proven to better reflect the changes in plasma volume under short-term periods of stress (<2 h) [27]. Table 1 shows a comparison of different physiological markers in serum or plasma at rest and after exercise.

### 3.2. Urine

The main function of the urinary system is to filter blood, eliminate wastes from the body, regulate blood volume and pressure, control levels of electrolytes and metabolites, and regulate blood pH. The end-product, urine, is a clear, light-yellow fluid that is produced in the process of cleaning the blood. Urine consists primarily of water (91–96%), with electrolytes (e.g., sodium, potassium, chloride, magnesium, calcium etc.), organic solutes (e.g., urea, creatinine, uric acid), trace amounts of enzymes, hormones, fatty acids and pigments [34]. A healthy adult excretes between 1000–1800 ml of urine per day. The amount and concentration of urine excreted vary with the level of exertion, the environment, the level of hydration, and dietary intake of salt and protein.

Two of the most common measures using urine are osmolality and urine-specific gravity (USG). Urine osmolality is a measure of the number of dissolved particles per unit of water in the urine, and this can vary between 50–1200 mOsmol/kg depending on an individual’s hydration status [35]. Urine-specific gravity is a measure of the ratio of the density of urine to the density of water, and this can vary between 1.002 to 1.030 [36]. The American College of Sports Medicine defines dehydration as USG ≥ 1.020 [37], and the higher the number, the more dehydrated the body [38]. Determination of urine osmolality is more important than specific gravity because it gives the most accurate measurement of total solute concentration and therefore provides the best measurement of the kidneys’ concentrating ability [39]. Athletes and their trainers must use USG in conjunction with alternative measures to ensure safe weight-loss methods in weight-classified sports [40].

Exercise increases the perfusion of active muscles, whereas the perfusion of body organs such as the kidneys may decrease by up to 25% of the resting value during strenuous exercise [41]. During exercise, the oxygen demand by muscles increases, and therefore blood flow is rerouted from internal organs, including the kidneys. This rerouting process leads to a decrease in glomerular filtration rate (GFR), renal plasma flow (RPF) and an increase in proteinuria and hematuria [42]. The changes in GFR and RPF during exercise are the result of increased renal sympathetic nerve activity and a rise in catecholamine secretion [41,43,44]. Changes in GFR, RPF and plasma antidiuretic hormone, in turn, affect the urine volume, osmolality and excretion of electrolytes (e.g., sodium, potassium, chloride and phosphate), urea nitrogen and lactic acid.

Exercise causes dehydration which affects the concentration of urinary biomarkers [4]. The level of dehydration depends on the type, intensity and duration of exercise, humidity and environmental temperature, as well as the hydration status of the individual [42]. During exercise, sweat glands have priority over the kidneys in their demand for salt. Therefore, urinary salt excretion is reduced due to the loss of salts in sweat [45]. Dehydration is common during exercise when sweat loss is not replaced, and this leads to a concentration effect in urine [4]. Numerous studies have documented the deleterious effects of dehydration on athletes and their performance during exercise [46,47]. Symptoms of dehydration can range from thirst, flushed skin, apathy and discomfort to dizziness, headaches, nausea, chills and vomiting when severe [48]. Comparison of blood serum sodium concentration ([Na+] > 145 mmol/L) and urine markers to measure dehydration showed that, while no athlete was dehydrated according to blood [Na^+^] measurement, 27%–55% of athletes were classified as dehydrated based on urine concentration limits [48]. This is because the urinary output is a response rather than a reflection of (tightly controlled) blood tonicity, and therefore healthcare professionals should be cautious when utilising urine indices to diagnose or monitor dehydration [49]. Table 2 shows a comparison of different physiological markers in urine at rest and after exercise.

Urine samples are categorised by the collection procedure used to obtain the specimen. The pooled collection sample, also known as a timed collection sample, is the most commonly used method to collect samples to measure creatinine, urea nitrogen, glucose, sodium, potassium, or analyte concentrations in urine over a specified length of time, usually 8 or 24 h. In order to determine the concentration and ratio of the analytes, precise recording of start and end times for the collection period are important [50]. However, this method should not be applied in understanding post-exercise changes, as the majority of the studied parameters show changes over time during exercise but return to normal values over the following 24 h [51]. Also, pooled urine collection is not practical with athletes during exercise; in order to examine post-exercise changes in urine biochemical parameters, single specimens are preferred [52].

**Table 2 nutrients-14-04685-t002:** Comparison of different urinary physiological markers at rest and after exercise.

Urinary Physiological Marker	Participants	Exercise Protocol [Reference]	At Rest	After Exercise	% Change	Urine Collection Method
Albumin	35 M	Cycling, 80% HR_max_, 30 min TT Cycling, 80% HR_max_, 120 min or 3% hypohydration achieved [11]	3.9 mg/mL	10.0 mg/mL 32.5 mg/mL	↑ 156 ↑ 733	Mid-stream urine before and after race
Calcium	21 M	Cross-country skiing, 70 km, 5.45 h TT [12]	6.6 mmol/L	2.3 mmol/L	↓ 66	Morning urine sample and spot urine after race
Chloride	21 M	Cross-country skiing, 70 km, 5.45 h TT [12]	131.4 mmol/L	115.6 mmol/L	↓ 12	Morning urine sample and spot urine after race
Creatinine	35 M	Cycling, 80% HR_max_, 30 min TT Cycling, 80% HR_max_, 120 min or 3% hypohydration achieved [11]	5.0 mmol/L	9.2 mmol/L 26.3 mmol/L	↑ 84 ↑ 426	Urine before and after race
	16 M	Running, 60 km [53]	0.2 mmol/L	0.2 mmol/L	↑ 59	Mid-stream urine before and after race
	21 M	Cross-country skiing, 70 km, 5.45 h TT [12]	19.3 mmol/L	28.0 mmol/L	↑ 45	Morning urine sample and spot urine after race
	24 M, F	Running, 120 km [54]	0.1 mmol/L	0.1 mmol/L	↑ 56	Mid-stream urine before and after race
Cystatin C	35 M	Cycling, 80% HR_max_, 30 min TT Cycling, 80% HR_max_, 120 min or 3% hypohydration achieved [11]	0.01 mg/L	0.03 mg/L 0.2 mg/L	↑ 200 ↑ 1400	Mid-stream urine before and after race
Glucose	35 M	Cycling, 80% HR_max_, 30 min TT Cycling, 80% HR_max_, 120 min or 3% hypohydration achieved [11]	0.1 mmol/L	0.2 mmol/L 0.5 mmol/L	↑ 55 ↑ 355	Mid-stream urine before and after race
Magnesium	21 M	Cross-country skiing, 70 km, 5.45 h TT [12]	8.1 mmol/L	2.7 mmol/L	↓ 67	Morning urine sample and spot urine after race
NGAL	14 M, F	Running, 0.8 km [55]	The change from baseline	12.8 ng/mL	---	Mid-stream urine before and after race
	35 M	Cycling, 80% HR_max_, 30 min TT Cycling, 80% HR_max_, 120 min or 3% hypohydration achieved [11]		1.0 ng/mL 11.1 ng/mL	---	Mid-stream urine before and after race
Osmolality	35 M	Cycling, 80% HR_max_, 30 min TT Cycling, 80% HR_max_, 120 min or 3% hypohydration achieved [11]		585.0 mOsm/kg 837.0 mOsm/kg	↑ 61 ↑ 130	Urine before and after race
Potassium	21 M	Cross-country skiing, 70 km, 5.45 h TT [12]	47.5 mmol/L	137.2 mmol/L	↑ 189	Morning urine sample and spot urine after race
Sodium	21 M	Cross-country skiing, 70 km, 5.45 h TT [12]	151.4 mmol/L	71.2 mmol/L	↓ 53	Morning urine sample and spot urine after race
Urea	21 M	Cross-country skiing, 70 km, 5.45 h TT [12]	496.8 mmol/L	230.8 mmol/L	↓ 54	Morning urine sample and spot urine after race
Volume	21 M	Cross-country skiing, 70 km, 5.45 h TT [12]	200.0 ml	135.0 ml	↓ 33	Morning urine sample and spot urine after race

M: males, F: females, TT: time trial, ↑: increase in % change, ↓: decrease in % change. NGAL: Neutrophil Gelatinase Associated Lipocalin.

### 3.3. Saliva

Saliva is a clear, slightly acidic mucoserous exocrine secretion that consists of 99% water and 1% electrolytes (e.g., sodium, potassium, calcium, magnesium, bicarbonate, phosphates), proteins, immunoglobulins, metabolites, enzymes, hormones and vitamins [56]. Whole saliva is formed primarily from the secretions of three major salivary glands (submandibular, parotid and sublingual glands), which together account for about 90% of the fluid production and electrolyte content; minor salivary glands (~600–1000 glands) account for the remaining 10%. Each salivary gland secretes a characteristic type of saliva with different ionic and protein characteristics [57]. Figure 2 shows the anatomical locations of the major and minor salivary glands. Table 3 lists the salivary glands, the composition of the saliva generated from these glands and the percentage contribution during unstimulated flow.

Proteomic analysis has been used to catalogue the salivary proteins generated from the different salivary glands and their ductal secretions. However, variations exist between reported results, possibly due to sample collection methods, storage conditions, sample integrity, analytical methods and the number of participants in the trials. This variation has hindered the use of saliva as a physiological and pathophysiological research tool and as a reliable fluid for disease diagnosis [58].

The main functions of saliva are to protect the oral tissues against bacteria, maintain pH, initiate the digestion of starch, enhance taste, provide antibacterial and antiviral activity and enable speech by lubricating the moving oral tissues [56,59]. Impaired salivary secretion (hyposalivation) has been shown to increase the risk of oral diseases, such as dental caries and oral candida infection [60].

The most abundant protein in human saliva is salivary α-amylase, an enzyme that is produced by the highly differentiated epithelial acinar cells of the exocrine salivary glands [61]. Salivary α-amylase cleaves large starch molecules into smaller chains of glucose called dextrins and maltose [62]. Glucose will then be generated from maltose via the action of disaccharide enzymes, such as maltase. In addition, salivary α-amylase plays a key role in the mucosal immunity of the oral cavity as it inhibits the adherence and growth of bacteria [63].

The constituents of saliva and rate of flow differ greatly between collection methods. Saliva collection can be divided into three categories: stimulated vs. unstimulated saliva, whole saliva vs. sampling from specific salivary glands, and those collected using absorbent materials vs. techniques based on passive drooling or spitting of saliva into collection tubes [64]. At rest, there is a small continuous flow of saliva without any exogenous or pharmacological stimulation. This unstimulated secretion is the baseline level that is present at all times in the form of a film that covers, moisturises, and lubricates the oral tissues [56]. In healthy adults, unstimulated secretion has a flow rate of about 0.4–0.5 mL/minute [65]. Unstimulated whole saliva is usually collected using the ‘draining’ or ‘drool’ method, where the subject’s head is tilted forward so that saliva moves towards the anterior region of the mouth and the pooled saliva drools into a sterile wide-mouthed container. Stimulated whole saliva is produced in response to a mechanical, gustatory, olfactory, or pharmacological stimulus and expectorated into a tube [66]; this accounts for 40–50% of daily salivary production [67]. The stimulated method of collection is undesirable for diagnostic applications as the foreign substances used to stimulate saliva production tend to alter the fluid pH and generally stimulate the water phase of saliva secretion, resulting in the dilution of proteins. Specifically, the contribution of the parotid gland increases from 20% to more than 50% of the total saliva [68].

Whole saliva collection is the most common and least invasive procedure [66]. Sampling saliva from individual salivary glands, such as parotid, submandibular, sublingual and minor glands, is possible using unique sampling methods, and the advantages and disadvantages of saliva sampling from specific salivary glands are discussed in detail by Bellagambi et al. [66]. Saliva collected using absorbent materials is performed by introducing a synthetic gauze sponge, pre-weighed swab or cotton pad into the mouth at the orifices of major salivary glands to collect stimulated or unstimulated saliva, which can then be extracted by centrifuging the absorbent material [64]. This method is easy to perform and advantageous when the salivary flow rate is low. A disadvantage of absorbent-based collection techniques is that stimulation of salivary flow cannot be completely excluded due to the presence of the mechanical stimulus in the mouth, even if participants are instructed not to chew on the material [69]. One study where saliva was collected in different areas of the mouth by placing absorbent cotton swabs in locations to specifically collect saliva from the three major glands yielded different values for salivary α -amylase from a sample of whole saliva [70].

Physical exercise is known to affect salivary secretion and induce changes in concentration, secretion rate, and composition among various salivary components, such as electrolytes, immunoglobulins, hormones, lactate and proteins [71,72]. In general, during intracellular dehydration, the salivary flow rate reduces, and saliva osmolality increases significantly. As saliva predominantly consists of water, the reduction of body fluids during dehydration leads to the salivary gland’s hypofunction [73]. Salivary immunoglobulin A (s-IgA) levels have been reported to depend on the intensity and duration of exercise, hydration status of athletes, saliva collection methods, diurnal variation and the method used to express s-IgA (i.e., whether s-IgA was reported as absolute concentration, secretion rate, ratio to total saliva protein, ratio to saliva albumin or ratio to saliva osmolality) [10,74,75,76,77]; which explains the variations in pre and post-exercise levels of s-IgA concentrations in Table 4, in trials carried out at varying levels of exercise intensity (maximal oxygen uptake or VO_2max_) as well as using different exercise protocols.

### 3.4. Assessment of Key Physiological Markers before and after Exercise

Many research studies have been conducted measuring physiological markers in blood, urine and saliva, at rest and after exercise. However, some markers such as creatinine, cystatin C, albumin, electrolytes and cortisol have been studied more consistently in relation to sports and exercise.

Table 1, Table 2 and Table 4 show the comparison of different blood physiological markers at rest and after exercise in blood, urine and saliva, respectively. The following section aims to introduce the above-mentioned markers, provide explanations for the results obtained across studies, and assess the utility of these markers in an exercise setting.

#### 3.4.1. Creatinine

Creatinine, a waste product generated during muscle metabolism, is produced from creatine, phosphocreatine and adenosine triphosphate (ATP) [78]. Creatinine is used in sports medicine to evaluate an athlete’s overall health, especially in events where hydro electrolytic balance is important [79]. Creatinine is a very useful physiological marker in kidney diagnostics as it is excreted only by a combination of glomerular filtration and secretion by the proximal tube. Creatinine is freely filtered by the glomerulus, and little or no reabsorption occurs in the tubules. This characteristic makes creatinine a useful tool to measure glomerular filtration, which in turn is a good indicator of kidney function [80].

After exercising, the serum creatinine concentration generally increases. This rise in serum creatinine concentration may be due to the release of creatinine from the working muscles, dehydration, and/or a reduction in renal blood flow, glomerular filtration rate and/or decrease in urinary excretion [81].

The amount of creatinine formed is proportional to muscle mass, resulting in differences in serum creatinine concentration related to sex and age [29]. Studies in males have shown increases in serum creatinine by 65% (21 males) [82] and 44% (70 males) [29] following exercise (Table 1). A study with 58 healthy volunteers (29 males and 29 females) who performed high-intensity interval resistance training found that serum creatinine increased significantly after exercise (24 h) in males but decreased steadily in females [31]. The distribution of lean and fat mass is sex-specific, with men generally having more lean muscle mass and females having more fat mass [83], and this explains the increase in serum creatinine after exercise in males due to their higher muscle mass.

The type of sport practised (training, aerobic/anaerobic metabolism, competitive season) and duration of exercise should also be taken into account when interpreting creatinine concentrations in athletes when exercising [11,84]. After exercise, serum creatinine increased by 30–65% among 37 subjects who performed long-duration exercises (e.g., races or marathons spanning around 240 min or 70 km) [30]. Similarly, urinary creatinine has also been shown to increase by as much as 45–426% following a 70 km cross-country ski race [82], 120 km marathon [54] or cycling at 80% HR_max_. [11]. However, another study with 58 subjects who performed short-duration (4 min) high-interval resistance training (squatting exercise) led to a 1% decrease in serum creatinine [31]. Therefore, the type of exercise and/or duration of exercise may affect the level of serum creatinine.

A number of studies have looked at the change in serum creatinine and urine creatinine due to exercise within the same trial [11,12,53]. Both serum and urine creatinine increased after exercise, with the level of increase varying between studies [11,12,30,53,54]. When comparing creatinine levels between the two body fluids in a study, some studies showed a greater percentage of increase in serum than in urine [12] and vice versa [11,53]. Variations in the percentage increase in creatinine due to exercise between studies are attributed to the different types of exercise protocols and sample sizes used in these studies.

#### 3.4.2. Cystatin C

Cystatin C is a 122-amino acid, 13-kDa protein that is a member of the family of cysteine proteinase inhibitors [85]. It is freely filtered at the glomerulus, but unlike creatinine, it is then reabsorbed and completely catabolised by the proximal renal tubules [86,87]. Serum cystatin C is inversely related to GFR, with high values indicating low GFR and vice versa.

Renal impairment, ranging from altered renal function to acute kidney injury, is common during exercise. Renal function is usually assessed in clinical practice using serum creatinine, urinary parameters, and equations to calculate eGFR (estimated glomerular filtration rate) [29]. eGFR is commonly calculated using the CKD-EPI formula and measures how much blood is filtered by the glomerulus in the kidney based on an individual’s age and sex [54]. However, serum cystatin C has been proven to be a better marker of eGFR than serum creatinine among athletes [29]. As discussed above, serum creatinine is affected by BMI, muscle mass and the type of exercise (training, aerobic/anaerobic metabolism, and competitive season) [29,84]. Some athletes (e.g., cyclists) have a serum creatinine level that is lower than the reference values due to their lean muscle mass, while some professional rugby players have serum creatinine levels above the reference values due to their higher muscle mass [86]. Nevertheless, both these cohorts of athletes with varying serum creatinine levels showed serum cystatin C concentrations within the normal range [88]. Therefore, cystatin C may be a better marker than serum creatinine to estimate GFR and renal function at rest and after exercise in athletes [29].

Many studies have examined the effect of exercise on cystatin C. As shown in Table 1, a 25–34% increase in serum cystatin C was seen after a marathon run, followed by a decrease in cystatin C levels, to pre-exercise values, within 24 h [29,32]. Comparisons of cystatin C levels after acute exercise (cycling, 80% HR_max_, 30 min) versus prolonged exercise (cycling, 80% HR_max_, 120 min or 3% hypohydration achieved) show comparable serum cystatin C levels after acute exercise, whereas prolonged exercise increased cystatin C levels [11]. This comparison indicates that the duration of exercise has an impact on the level of cystatin C increase that is due to exercise.

When the effect of the exercise duration on urine cystatin C levels was studied, a significant increase in urine cystatin C was observed, with higher values after prolonged (150 min) compared to acute (30 min) exercise (Table 2). However, more studies are needed to better understand the impact of exercise on urine cystatin C, but at this stage, it appears to be a very sensitive marker of proximal tubule dysfunction after exercise [89].

#### 3.4.3. Albumin

Albumin, a globular protein (66–69 kDA), is essential for the transportation of substances such as calcium, bilirubin and progesterone in the blood and for the maintenance of osmotic balance [90]. Albumin correlates with the protein level in the body, where strenuous exercise increases the excretion of albumin, and low protein intake reduces albumin synthesis rates [91,92,93].

Serum albumin increased by 10–13% after running for approximately 4 h [29,30] due to dehydration and decreased renal perfusion (Table 1) [94]. Several studies have shown that exercise causes an increase in urine albumin [95,96,97]. A study by Bongers et al. showed that urine albumin increased by 156% and 733% after 30 and 150 min, respectively, of cycling at 80% HR_max_ (Table 2) [11]. This indicates that although all forms of exercise may cause an increase in albumin excretion, the magnitude of the increase varies depending on the duration and intensity of exercise [98].

The cause of exercise-induced proteinuria is unclear but may be due to either the plasma concentration of angiotensin II increasing during exercise, leading to filtration of protein through the glomerular membrane, or strenuous exercise increasing the sympathetic nervous activity and catecholamine levels in the blood, which leads to greater permeability of the glomerular capillary membrane [95,99].

Maintaining an optimal protein balance is essential for athletes for training and quick recovery and is also an important tool for tracking their health and kidney status. A low albumin level in athletes is a marker of a lack of protein intake in the absence of disease. But, when protein intake appears to be adequate, albumin levels may suggest other health concerns for the athlete [100,101]. Proteinuria or albuminuria, a pathological condition where protein albumin is abnormally present in the urine, is a sign of kidney disease and has been linked to strenuous exercise [101].

#### 3.4.4. Electrolytes

Electrolytes must be present at optimal concentrations in order to maintain fluid balance, muscle contraction and neural activity when exercising. Large amounts of electrolytes, particularly sodium, are lost in sweat [102], which is hypotonic relative to plasma, so sweat-mediated hypohydration will act to increase plasma osmolality but decrease plasma volume [103]. As shown in Table 1, some studies showed an increase in serum sodium concentrations following exercise [104,105,106], while others indicated no change or a decrease in sodium concentrations immediately after a race or 24 h later [30]. An increase in serum sodium concentrations is noted when there is a greater loss of water than sodium during exercise. A decrease in serum sodium concentrations is often noted in ultramarathons and Ironman triathlons when marathon runners develop symptomatic hyponatremia after drinking large quantities of water during a marathon run under warm conditions [107]. Maintaining blood sodium concentrations is critical for an athlete’s health, with the effects of exercise-associated hyponatraemia being a major source of concern. Potassium ions, on the other hand, are released from contracting muscles during exercise, leading to an increase in plasma potassium concentrations and a decrease in intracellular potassium concentrations [108].

Within the urinary system, aldosterone, a mineralocorticoid hormone, plays an important role in the regulation of electrolytes by acting on the distal renal tubes [109]. It promotes the absorption of sodium ions from the renal tubules and, together with the reduction in glomerular filtration rate (to balance the rise in sodium loss through sweat), leads to a decrease in urinary sodium levels [107]. Aldosterone also increases the urine extraction of potassium ions leading to an increase in urinary potassium levels (Table 2).

In the field of sports and exercise, knowledge of why electrolytes are important may be useful for athletes and encourage them to better track their fluid balance to replace the salts lost and prevent the onset of cramps or fatigue due to this loss.

#### 3.4.5. Cortisol

Cortisol is a steroid hormone that plays an important role in the body’s behavioural and physiological responses to stressful conditions such as exercise and extreme temperatures [9]. When exercising, the hypothalamus secretes corticotrophin-releasing hormone (CRH), which activates the anterior pituitary to release adrenocorticotropic hormone (ACTH), which in turn stimulates the adrenal cortex to release cortisol [110]. Cortisol aids in exercise capacity and recovery by breaking down proteins and triglycerides in adipose tissue to be hydrolysed into free fatty acids and glycerol [110,111]. Free fatty acids released into the blood are transported and taken up by other tissues for ATP regeneration, while glycerol enters the glycolytic pathway [112]. Approximately 80% of cortisol is transported bound to the specific carrier protein cortisol binding globulin, 10% is bound to albumin, and only 10% is free (unbound) and biologically active [113].

The type, intensity and duration of exercise, as well as the training status of the athlete, are known to influence serum cortisol levels [114]. In general, low-intensity exercise is associated with no change or even a decrease in the concentration of cortisol, while exercise performed at above 60% VO_2max_ is associated with an acute increase in serum cortisol, as shown in Table 1 [115].

Salivary cortisol levels have been used as a measure to reflect stress due to exercise. Cortisol, with its low molecular weight, can easily enter body fluids by passive diffusion, thus making it possible to measure free cortisol levels in all fluids, including saliva [110]. Salivary cortisol levels linearly relate to exercise intensity and duration [116], with cortisol levels increasing significantly for both serum and saliva only in response to high-intensity exercise [9]. It is important to note that cortisol exhibits a circadian rhythm, with peak concentrations in the morning, around the commencement of diurnal activity, and reduced concentrations in the evening and overnight, and therefore caution is needed when interpreting results to ensure sample collection times occur at the same time of day or night [117]. Even though cortisol levels increase in relation to the intensity of exercise, continuous excessive stress over time decreases the cortisol response to exercise, potentially resulting in a disorder called ‘overtraining syndrome’, which in turn negatively impacts the athlete’s performance [118,119,120]. Therefore, assessment of cortisol secretion along with circadian rhythms is helpful for diagnosis and as a tool for the prevention of overtraining syndrome in athletes [121].

The use of salivary cortisol measures offers several advantages, including measurement at rest and during exercise, as salivary cortisol levels are thought to have a steady and predictable relationship to free and total cortisol levels [122]. Salivary cortisol sample collection is easy, non-invasive and can be undertaken while exercising using a Salivette polyester swab device that does not adsorb steroids [123]. Moreover, salivary levels of cortisol are not affected by variations in flow rate [124]. Even though multiple studies have shown that salivary cortisol levels mirror serum cortisol levels during exercise [124,125,126,127], there is still controversy surrounding the use of salivary cortisol levels instead of serum levels. One study comparing salivary and serum cortisol levels reported that the correlation between the two varied depending on the method of saliva collection, with saliva collected using Salivettes a better predictor for free serum cortisol than passive drooling [128]. Since cortisol moves passively from blood into saliva, there will be a delayed response in the salivary cortisol levels before correctly reflecting the level in the blood [9]. Also, the effect of training on resting salivary cortisol levels is equivocal-some studies have shown training increases resting cortisol levels [9,76] while others indicate a decrease [56].

#### 3.4.6. Other Markers

Other markers measured in blood, urine and/or saliva include urea, IgA and glucose (Table 1, Table 2 and Table 4). After exercise, urea levels increase in blood and decrease in urine; both these effects can be attributed to increased dehydration and decreased renal perfusion following exercise [129]. Salivary IgA levels increase during training and then reduce below the pre-exercise level after training [75,130,131,132]. This change could be due to a reduction in the amount of saliva secreted during exercise, which would lead to more concentrated saliva. Urine glucose levels have been found to increase after acute and prolonged exercise, according to several studies [133,134], which suggests that the proximal reabsorption of glucose deteriorates as a consequence of kidney stress. Understanding these marker levels is relevant in the field of sports and exercise as it indicates an athlete’s hydration status and immunity status, and this, in turn, affects their performance and recovery.

## 4. Limitations and Future Research

Physiological markers are useful tools to assess and monitor health, training status and performance. Many studies have investigated the impact of exercise on body fluids; however, the results are frequently not consistent across studies. While the direction of change (increase or decrease) of physiological markers due to exercise is mostly similar, the level of change differs between studies. Some of the variations in results may be due to differences in study design, different equipment or statistical formula used, variability in time intervals for after-race sampling, sample population and size, fitness levels, nutritional intake and environmental factors during exercise.

When comparing changes in analyte levels between different body fluids, a similar direction of change can be seen due to exercise among most physiological markers, although again, the percentage change varies. This is to be expected as time delays are likely to occur due to the transport of analytes from blood to other body fluids such as saliva and particularly urine. Urine analyte concentrations will be more susceptible to changes due to water intake and urine volume, with urine concentration reflecting urine collected in the bladder over time which could be a period of hours, as opposed to blood, which reflects the current physiological status [14,135]. In theory, urine is an ideal clinical specimen for analysis in sports medicine as it is excreted in large quantities. The collection does not require invasive methods and can be collected in a range of field settings. However, urine is a pooled body fluid collected and stored over a period of time. It is slow in detecting changes in hydration status during periods of rapid body fluid turnover [136] and shows considerable intra-individual as well as inter-individual variation [137,138]. Saliva can also be collected non-invasively. However, saliva also shows large variability in flow rate during low levels of hydration [5] and variation in results between participants due to the interference of food and beverage consumption, oral hygiene routines, dental erosions and circadian rhythm [6]. Table 5 lists the pros and cons of using blood, urine and salivary biomarkers in an exercise setting. In order to overcome the variations noted between these body fluids, it would be advisable to investigate electrolytes, osmolality and IgA levels in all three body fluids simultaneously, thereby enabling a complete comparison of these levels and producing results that could be used to improve athletes’ performances. 

Most studies have used only males or males and females combined, and there are no studies reporting female-only data. As discussed previously, the results of physiological markers are sex-dependent. Therefore, it would be useful to also study these markers in female-only populations to gain a better understanding of how the levels of these markers change in relation to exercise and performance. To better understand the use of other body fluids in place of blood, more work needs to be performed to understand if there are correlations over time between analytes in the different fluids. Also, future research needs to be undertaken to understand the best collection procedures for each physiological marker. Further research conducted using standardised methods will enable athletes and practitioners to fully understand the effect of exercise on physiological markers and use that information to facilitate improvements in performance and recovery.

Renal impairment, ranging from altered renal function to acute kidney injury, is common during exercise. Renal function is usually assessed in clinical practice using serum creatinine, urinary parameters, and equations to calculate eGFR. In athletes, serum cystatin C has been proven to be a better marker of eGFR than serum creatinine [29]. In addition, serum creatinine is affected by BMI, muscle mass and the type of exercise (training, aerobic/anaerobic metabolism, and competitive season) [29,84]. Athletes with lean muscle mass (e.g., cyclists) or higher muscle mass [88] have serum creatinine levels that are lower or higher than reference values, respectively. Yet both sets of athletes had serum cystatin C concentrations within the normal range [88]. Therefore, in athletes, serum cystatin C may be a better marker than serum creatinine to estimate GFR and renal function at rest and after exercise in athletes [29].

## 5. Conclusions

Compared to a rested state, exercise poses a substantial increase in demand on the body and disturbs the body’s homeostasis, causing a shift in hemodynamic and metabolic processes. When the effect of exercise on physiological markers such as albumin, cortisol, creatinine, cystatin c, potassium, calcium, chloride and magnesium was reviewed across similar studies and compared among various body fluids such as blood, urine and saliva, observed changes were generally in the same direction. The degree of change varied, however, possibly due to the type and duration of exercise in the study design, sample population and size, fitness levels and/or nutritional intake. The finding of a correlation in the direction of changes in the levels of physiological markers shift in different body fluids as a result of exercise is very promising. It means that with further research, using standardised methods specific for the type of body fluid, other body fluids such as urine and saliva, which are simple, non-invasive and inexpensive to collect, may be able to be used more often to gain insight when studying the effects of exercise on aspects such as performance.

## Figures and Tables

**Figure 1 nutrients-14-04685-f001:**
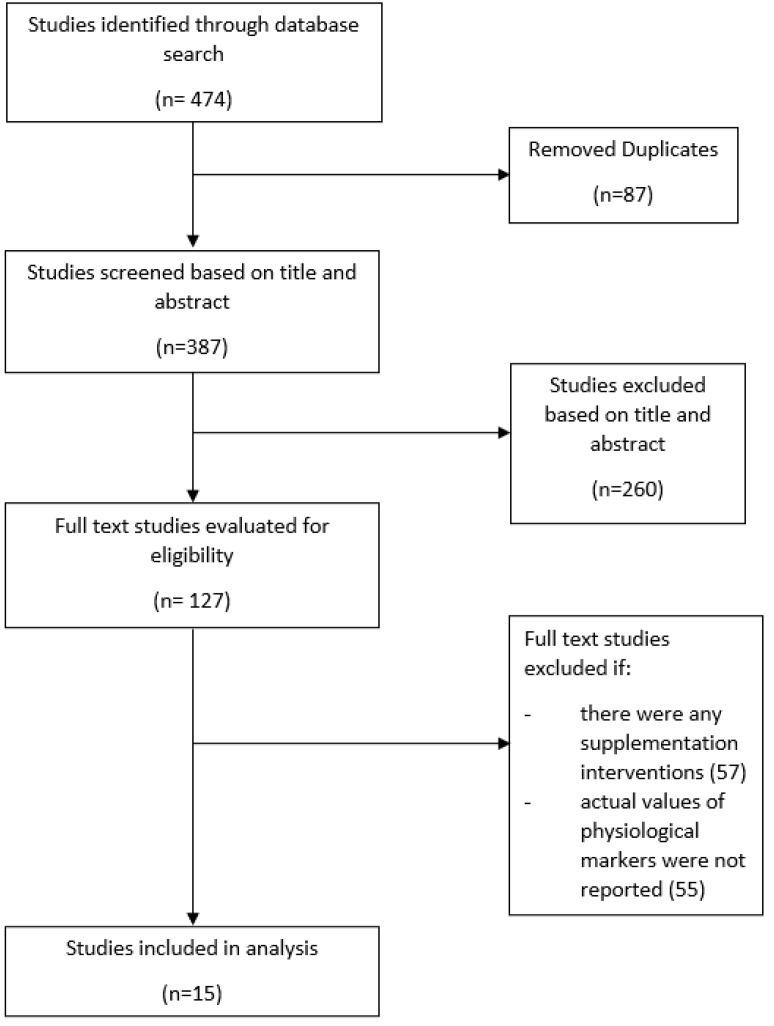
PRISMA flow diagram of the search procedure used for this narrative review.

**Figure 2 nutrients-14-04685-f002:**
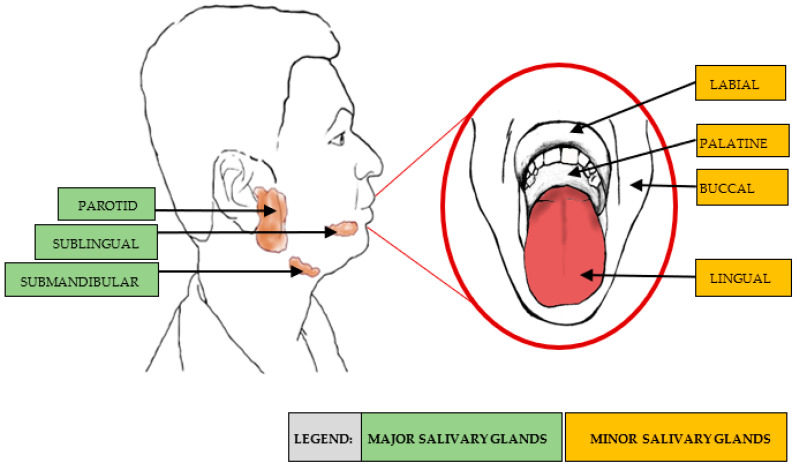
Locations of the major and minor salivary glands.

**Table 1 nutrients-14-04685-t001:** Comparison of different blood physiological markers in serum or plasma at rest and after exercise.

Physiological Marker	Participants	Exercise Protocol [Reference]	At Rest	After Exercise	% Change	Blood Matrix
Albumin	70 M	Running, 222.6 min TT [29]	44.0 g/l	48.2 g/l	↑ 10	Serum
	37 M, F	Running, 240 min TT [30]	40.0 g/l	45.0 g/l	↑ 13	Serum
Calcium	21 M	Cross-country skiing, 70 km, 5.45 h TT [12]	2.4 mmol/L	2.4 mmol/L	↓ 0.4	Serum
Chloride	21 M	Cross-country skiing, 70 km, 5.45 h TT [12]	106.5 mmol/L	106.2 mmol/L	↓ 0.3	Serum
	37 M, F	Running, 240 min [30]	105.7 mmol/L	101.0 mmol/L	↓ 4	Serum
Cortisol	12 M	Cycling, 40% of VO_2max_ Cycling, 60% of VO_2max_ Cycling, 80% of VO_2max_ [9]	394.5 nmol/l 386.2 nmol/l 336.6 nmol/l	380.7 nmol/l 446.9 nmol/l 565.5 nmol/l	↓ 4 ↑ 16 ↑ 68	Serum
Creatinine	70 M	Running, 222.6 min TT [29]	0.1 mmol/L	0.1 mmol/L	↑ 44	Serum
	37 M, F	Running, 240 min [30]	0.1 mmol/L	0.1 mmol/L	↑ 30	Serum
	21 M	Cross-country skiing, 70 km, 5.45 h TT [12]	0.1 mmol/L	0.2 mmol/L	↑ 65	Serum
	58 M, F	Squat exercise, 4 min TT [31]	0.1 mmol/L	0.08 mmol/L	↓ 1	Serum
Cystatin C	70 M	Running, 222.6 min TT [29]	0.7 mg/l	1.0 mg/l	↑ 34	Serum
	25 M, F	Running, 256.2 min TT [32]	0.8 mg/l	1.0 mg/l	↑ 25	Serum
	167 M, F	Running, 263.0 min TT [33]	0.7 mg/l	0.9 mg/l	↑ 25	Serum
Hemoglobin	35 M	Cycling, 80% HR_max_, 30 min TT Cycling, 80% HR_max_, 120 min or 3% hypohydration achieved [11]	9.2 mmol/L	9.6 mmol/L 9.7 mmol/L	↑ 4 ↑ 5	Plasma
	37 M, F	Running, 240 min TT [30]	9.2 mmol/L	9.4 mmol/L	↑ 2	Serum
Magnesium	21 M	Cross-country skiing, 70 km, 5.45 h TT [12]	0.9 mmol/L	0.8 mmol/L	↓ 9	Serum
	37 M, F	Running, 240 min TT [30]	1.7 mmol/L	1.5 mmol/L	↓ 12	Serum
Potassium	37 M, F	Cross-country skiing, 70 km, 5.45 h TT [12]	4.4 mmol/L	4.7 mmol/L	↑ 8	Serum
Sodium	35 M	Cycling, 80% HR_max_, 30 min TT Cycling, 80% HR_max_, 120 min or 3% hypohydration achieved [11]	142.2 mmol/L	141.9 mmol/L 144.0 mmol/L	↓ 0.2 ↑ 1	Serum
	21 M	Cross-country skiing, 70 km, 5.45 h TT [12]	138.7 mmol/L	140.1 mmol/L	↑ 1	Serum
	25 M, F	Running, 240 min TT [30]	142.4 mmol/L	141.5 mmol/L (4 h) 139.0 mmol/L (24 h)	↓ 0.6 ↓ 2	Serum
Urea	21 M	Cross-country skiing, 70 km, 5.45 h TT [12]	5.7 mmol/L	9.0 mmol/L	↑ 56	Serum

M: males, F: females, TT: time trial, ↑: increase in % change, ↓: decrease in % change.

**Table 3 nutrients-14-04685-t003:** The different types of salivary glands, their location, composition and contribution to total saliva production.

Salivary Gland	Location	Composition	Contribution during Unstimulated Flow
**Parotid**	Side of the face, below and in front of each ear	Serous saliva: a watery secretion rich in enzymes, e.g., amylase and proline-rich proteins (PRP)	20%
**Submandibular**	Near the inner side of the lower jawbone, in front of the sternomastoid muscle	Mixed secretion that is both serous and mucous	65%
**Sublingual**	Directly under the mucous membrane covering the floor of the mouth beneath the tongue	Mucous saliva: a viscous secretion containing no enzymes and large amounts of mucus	7–8%
**Minor salivary glands**	Spread throughout the submucosa of the sinonasal cavity, oral cavity, pharynx, larynx, trachea, lungs, and middle ear cavity	Secretion that is dependent on location; includes pure mucous or serous, or mixed secretion	10%

**Table 4 nutrients-14-04685-t004:** Comparison of different salivary physiological markers at rest and after exercise.

Salivary Physiological Marker	Participants	Exercise Protocol [Reference]	At Rest	After Exercise	% Change	Salivary Collection Method
**Cortisol**	98 M, F	Running, 42.2 km, 268.2 min TT [76]	830.0 nmol/l	1035.0 nmol/l	↑ 25	Unstimulated whole saliva
	12 M	Cycling, 40% of VO_2max_ Cycling, 60% of VO_2max_ Cycling, 80% of VO_2max_ [9]	7.4 nmol/l 5.2 nmol/l 4.7 nmol/l	5.9 nmol/l 8.1 nmol/l 9.6 nmol/l	↓ 20 ↑ 55 ↑ 105	Mix method (passive drooling and if saliva secretion needed to be stimulated, subjects chewed on paraffin film)
	10 M	Cycling, 50% of VO_2max_ Cycling, 75% of VO_2max_ [24]	13.7 nmol/l 13.7 nmol/l	13.1 nmol/l 14.6 nmol/l	↓ 4 ↑ 7	Unstimulated whole saliva
**Glucose**	98 M, F	Running, 42.2 km, 268.2 min TT [76]	5.5 nmol/l	5.0 nmol/l	↓ 9	Unstimulated whole saliva
**IgA** **concentration**	98 M, F	Running, 42.2 km, 268.2 min TT [76]	399.0 mg/l	429.0 mg/l	↑ 8	Unstimulated whole saliva
	18 M	Cycling, 55% VO_2max_ Cycling, 80% VO_2max_ [10]	91.0 mg/l 85.0 mg/l	234.0 mg/l 295.0 mg/l	↑ 157 ↑ 247	Salivette swab
	22 M, F	Swimming, 12 weeks of 600–1500 min pool training + 300 min dry land training per week [56]	55.3 mg/l	37.1 mg/l	↓ 33	Unstimulated whole saliva
**Salivary** **secretion rate**	98 M, F	Running, 42.2 km, 268.2 min TT [76]	0.9 ml/min	0.6 ml/min	↓ 38	Unstimulated whole saliva
**Salivary** **amylase** **secretion rate**	10 M	Cycling, 50% of VO_2max_ Cycling, 75% of VO_2max_ [24]	226.0 U/min 150.0 U/min	200.0 U/min 264.0 U/min	↓ 12 ↑ 76	Unstimulated whole saliva

M: males, F: females, TT: time trial, ↑: increase in % change, ↓: decrease in % change.

**Table 5 nutrients-14-04685-t005:** Pros and cons of the blood, urine and salivary biomarkers in an exercise setting, and highlighting the ideal marker to measure hydration status within each body fluid.

	Pros	Cons
**Blood**	- Does not differ substantially between studies - Provides a point-of-care measurement to assess optimal training and recovery levels [139] - Serum cystatin C is a better biomarker to estimate GFR and renal function	- Invasive procedure requiring a medical technician - Difficult to obtain samples during exercise
**Urine**	- Non-invasive method - Of urine markers, USG provides the most accurate measurement of kidneys’ concentrating ability over time.	- Difficult to obtain samples during exercise - USG is still not a true reflection of dehydration when compared to blood markers
**Saliva**	- Non-invasive method - Saliva osmolality is an effective hydration assessment marker during active heat exposure [140]	- Highly variable between participants and collection methods

## Data Availability

Not applicable.

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
