# Peer review of "Assessment of Changes in Physiological Markers in Different Body Fluids at Rest and after Exercise"

_nutrients, 2022, doi:10.3390/nu14214685_

Round 1
Reviewer 1 Report
· Lines 49-51: The following sentence should be cited by more references:
“In the field of sports and exercise medicine, the use of physiological markers has become a popular method for monitoring an athlete’s training response, as the body’s physiology is significantly different at rest and after exercise [5].”
· Lines 52-54: Provide a reference(s) for the following sentence:
“In sports and exercise, body fluids such as urine and saliva are preferred, as the collection is simple, non-invasive, cheap and does not require a specialised technician.”
· Lines 347-348: The following sentence should be cited by several references:
“Both serum and urine creatinine increased after exercise, with the level of increase varying between studies.”
Author Response
We thank the reviewers for their constructive comments and hope the changes we have made are acceptable.
Reviewer 1 Comments
Lines 49-51: The following sentence should be cited by more references:
“In the field of sports and exercise medicine, the use of physiological markers has become a popular method for monitoring an athlete’s training response, as the body’s physiology is significantly different
References have been added as requested. Line 62
Lines 52-54: Provide a reference(s) for the following sentence:
“In sports and exercise, body fluids such as urine and saliva are preferred, as the collection is simple, non-invasive, cheap and does not require a specialised technician.”
References have been added as requested. Line 65
Lines 347-348: The following sentence should be cited by several references:
“Both serum and urine creatinine increased after exercise, with the level of increase varying between studies.”
References have been added as requested. Lines 376-377
Reviewer 2 Report
Thank you very much for allowing me to review the review article entitled "Assessment of changes in physiological markers in different body fluids at rest and after exercise" (nutrients-2008756), which is presented for the "Sport Nutrition" section of this journal.
The aims of this review are to compare the results obtained across studies measuring physiological markers pre and post exercise and suggest possible reasons for the variations in results obtained, with a focus on the population and sample collection or methodology used. Also, where possible, the change in (physiological marker) level due to the same or similar exercise protocol, between different body fluids will be compared.
Major Comments
The abstract must present the background, the objective, the methodology and the main results and conclusions of the study, therefore, I suggest the authors rewrite the abstract since it only raises the interest of the topic but does not give the reader a complete idea of the work done.
The introduction raises the issue but should also include the consequence that the instability of the fluids can have or the negative effect to give greater interest to the subject.
Material and methods: a narrative and comprehensive review of the literature (2020_2021) is carried out. it is well explained, but I suggest that a graph be incorporated according to prism that allows a better understanding of the selection of articles made. It should be explained why table 3 mentioned in this section does not appear.
The results of the discussion of the topic are presented simultaneously. Since it is a narrative review, it is usually done like this, but the section should be headed with the title of results and discussion.
The tables are very well elaborated and allow a better understanding of the subject. Especially interesting in table 5.
I considered that the review carried out will allow the most effective methodological approach in future studies in the selection of the most appropriate fluid monitoring for each type of exercise.
Minor comments:
Text revision, such as: in abstract:
Absorption: absorptions
has: have…
Author Response
We thank the reviewers for their constructive comments and hope the changes we have made are acceptable.
Reviewer 2
Thank you very much for allowing me to review the review article entitled "Assessment of changes in physiological markers in different body fluids at rest and after exercise" (nutrients-2008756), which is presented for the "Sport Nutrition" section of this journal.
The aims of this review are to compare the results obtained across studies measuring physiological markers pre and post exercise and suggest possible reasons for the variations in results obtained, with a focus on the population and sample collection or methodology used. Also, where possible, the change in (physiological marker) level due to the same or similar exercise protocol, between different body fluids will be compared.
Major Comments
The abstract must present the background, the objective, the methodology and the main results and conclusions of the study, therefore, I suggest the authors rewrite the abstract since it only raises the interest of the topic but does not give the reader a complete idea of the work done.
Thank you for your comments, we have rewritten the abstract (shown below) to better reflect the content including methods use, results and conclusions. Lines …23-33
Literature searches were conducted using PRISMA guidelines for keywords, such as exercise, physical activity, serum, sweat, urine, biomarkers, resulting in analysis of 15 studies for this review paper. When comparing the effects of exercise on physiological markers across different body fluids (blood, urine, and saliva) the changes detected were generally in the same direction. However, the extent of the change varied, potentially as a result of the type and duration of exercise, the sample population and subject numbers, fitness levels, and/or dietary intake. In addition, none of the studies used solely female participants; instead, including males only or both male and female subjects together. Results of some physiological markers are sex-dependent. Therefore to better understand how the levels of these biomarkers change in relation to exercise and performance, the sex of the participants should also be taken into consideration.
The introduction raises the issue but should also include the consequence that the instability of the fluids can have or the negative effect to give greater interest to the subject.
Thank you for the comment we have added the following to the introduction to indicate some of the limitations of different body fluids. Lines 49-54.
However, the composition of some body fluids are useful only at certain time points, which limits their usefulness; for example measures of urine specific gravity are a useful measure of hydration status prior to exercise but not post exercise [4]. Saliva shows a large variability in flow rate during low levels of hydration [5], as well as variation in results due to interference of food and beverage consumption, oral hygiene routines, dental erosions and circadian rhythm [6].
Material and methods: a narrative and comprehensive review of the literature (2020_2021) is carried out. it is well explained, but I suggest that a graph be incorporated according to prism that allows a better understanding of the selection of articles made. It should be explained why table 3 mentioned in this section does not appear.
A figure showing the process of article selection has been added (Fig 1). Lines 88-92
Table 3 has been added to the methods section Line 88.
The results of the discussion of the topic are presented simultaneously. Since it is a narrative review, it is usually done like this, but the section should be headed with the title of results and discussion.
The title of the section has been amended to Results and Discussion as suggested. Line 94
The tables are very well elaborated and allow a better understanding of the subject. Especially interesting in table 5. Thank you, we appreciate your comment.
I considered that the review carried out will allow the most effective methodological approach in future studies in the selection of the most appropriate fluid monitoring for each type of exercise.
Minor comments:
Text revision, such as: in abstract:
Absorption: absorptions This sentence was deleted when the abstract was rewritten.
has: have…This has been changed Line 17